# Genomic DNA transposition induced by human PGBD5

**Anton G Henssen[1], Elizabeth Henaff[2], Eileen Jiang[1], Amy R Eisenberg[1], Julianne R Carson[1], Camila M Villasante[1], Mondira Ray[1], Eric Still[1], Melissa Burns[3], Jorge Gandara[2], Cedric Feschotte[4], Christopher E Mason[2], Alex Kentsis[1,5,6]***

[1]Molecular Pharmacology Program, Sloan Kettering Institute, Memorial Sloan Kettering Cancer Center, New York, United States; [2]Institute for Computational Biomedicine, Weill Cornell Medical College, New York, United States; [3]Boston Children's Hospital, Harvard Medical School, Boston, United States; [4]Department of Human Genetics, University of Utah School of Medicine, Salt Lake City, United States; [5]Department of Pediatrics, Memorial Sloan Kettering Cancer Center, New York, United States; [6]Weill Cornell Medical College, Cornell University, New York, United States

**Abstract** Transposons are mobile genetic elements that are found in nearly all organisms, including humans. Mobilization of DNA transposons by transposase enzymes can cause genomic rearrangements, but our knowledge of human genes derived from transposases is limited. In this study, we find that the protein encoded by human *PGBD5*, the most evolutionarily conserved transposable element-derived gene in vertebrates, can induce stereotypical cut-and-paste DNA transposition in human cells. Genomic integration activity of PGBD5 requires distinct aspartic acid residues in its transposase domain, and specific DNA sequences containing inverted terminal repeats with similarity to *piggyBac* transposons. DNA transposition catalyzed by PGBD5 in human cells occurs genome-wide, with precise transposon excision and preference for insertion at TTAA sites. The apparent conservation of DNA transposition activity by PGBD5 suggests that genomic remodeling contributes to its biological function.

***For correspondence:** kentsisresearchgroup@gmail.com

**Competing interests:** The authors declare that no competing interests exist.

## Introduction

Transposons are genetic elements that are found in nearly all living organisms (*Feschotte and Pritham, 2007*). They can contribute to the developmental and adaptive regulation of gene expression and are a major source of genetic variation that drives genome evolution (*Cordaux and Batzer, 2009*). In humans and other mammals, they comprise about half of the nuclear genome (*Smit, 1999*). The majority of primate-specific sequences that regulate gene expression are derived from transposons (*Jacques et al., 2013*), and transposons are a major source of structural genetic variation in human populations (*Stewart et al., 2011*).

While the majority of genes that encode transposase enzymes tend to become catalytically inactive and their transposon substrates tend to become immobile in the course of organismal evolution, some can maintain their transposition activities (*Liu et al., 2007*; *Munoz-Lopez and Garcia-Perez, 2010*). In humans, at least one hundred L1 long interspersed repeated sequences (LINEs) actively transpose in human genomes and induce structural variation (*Kazazian, 2004*), including somatic rearrangements in neurons that may contribute to neuronal plasticity (*Erwin et al., 2014*). The human *Transib*-like transposase RAG1 catalyzes somatic recombination of the V(D)J receptor genes in lymphocytes and is essential for adaptive immunity (*Hiom et al., 1998*). The *Mariner*-derived transposase SETMAR

**eLife digest** Transposons are mobile genetic elements that can be cut out of and inserted into DNA. They are present in most living things and make up almost half of the human genome. Transposons help to rearrange and increase the variety of DNA sequences, which can drive evolution and regulate the expression of genes. Enzymes called transposases help to move transposons, but very few genes that encode these enzymes have been studied in humans.

PiggyBac transposase—which was first discovered in the cabbage looper moth—helps to move transposons of the *piggyBac* family. Humans and many other animals have genes that encode similar enzymes. In particular, the gene that encodes the human PGBD5 transposase is expressed in the developing embryo and particular areas of the brain and is highly similar to genes found in other vertebrate animals. These intriguing features prompted Henssen et al. to investigate PGBD5.

The experiments reveal that PGBD5 is able to move *piggyBac*-like transposons in human cells and insert them into sites that contain similar DNA sequences that are preferred by other PiggyBac transposases. Henssen et al. compared human PGBD5 to the piggyBac transposases from other organisms, including insects, bats, and frogs. They found that PGBD5 is deeply conserved among vertebrate organisms, and is distinct from other piggyBac transposases.

These findings suggest that PGBD5 is indeed a fully working piggyBac transposase. Further work is needed to understand what portions of the human genome may be rearranged by PGBD5, and how this may contribute to human brain development or disease.

functions in single-stranded DNA resection during DNA repair and replication in human cells and can catalyze DNA transposition in vitro (*Liu et al., 2007*; *Shaheen et al., 2010*).

Among transposase enzymes that can catalyze excision and insertion of transposon sequences, DNA transposases are distinct in their dependence only on the availability of competent genomic substrates and cellular repair enzymes that ligate and repair excision sites, as compared to retrotransposons, which require transcription of the mobilized sequences (*Berg and Howe, 1989*). Most DNA transposases utilize an RNase H-like domain with three aspartate or glutamate residues (so-called DDD or DDE motif) that catalyze magnesium-dependent hydrolysis of phosphodiester bonds and strand exchange (*Keith et al., 2008*; *Mitra et al., 2008*; *Dyda et al., 2012*). The IS4 transposase family, which includes piggyBac transposases, is additionally distinguished by precise excisions without modifications of the transposon flanking sequences (*De Palmenaer et al., 2008*). The piggyBac transposase and its transposon were originally identified as an insertion in lepidopteran *Trichoplusia ni* cells (*Fraser et al., 1985*). The *piggyBac* transposon consists of 13-bp inverted terminal repeats (ITRs) and 19-bp subterminal inverted repeats located 3 and 31 base pairs from the 5′ and 3′ ITRs, respectively (*Elick et al., 1997*). PiggyBac transposase can mobilize a variety of ITR-flanked sequences and has a preference for integration at TTAA target sites in the host genome (*Fraser et al., 1983*; *Beames and Summers, 1990*; *Wang and Fraser, 1993*; *Fraser et al., 1995*; *Elick et al., 1997*; *Handler et al., 1998*; *Mitra et al., 2008*).

Members of the piggyBac superfamily of transposons have colonized a wide range of organisms (*Sarkar et al., 2003*), including a recent and likely ongoing invasion of the bat *Myotis lucifugus* (*Mitra et al., 2013*). The human genome contains five paralogous genes derived from *piggyBac* transposases, *PGBD1-5* (*Smit and Riggs, 1996*; *Sarkar et al., 2003*). *PGBD1* and *PGBD2* invaded the common mammalian ancestor, and *PGBD3* and *PGBD4* are restricted to primates, but are all contained as single coding exons, fused in frame with endogenous host genes, such as the Cockayne syndrome B gene (CSB)-PGBD3 fusion (*Sarkar et al., 2003*; *Newman et al., 2008*). Thus far, only the function of *PGBD3* has been investigated. CSB-PGBD3 is capable of binding DNA, including endogenous *piggyBac*-like transposons in the human genome, but has no known catalytic activity, though biochemical and genetic evidence indicates that it may participate in DNA damage response (*Bailey et al., 2012*; *Gray et al., 2012*).

*PGBD5* is distinct from other human *piggyBac*-derived genes by having been domesticated much earlier in vertebrate evolution approximately 500 million years (My) ago, in the common ancestor of cephalochordates and vertebrates (*Sarkar et al., 2003*; *Pavelitz et al., 2013*). *PGBD5* is transcribed as a multi-intronic but non-chimeric transcript predicted to encode a full-length transposase (*Pavelitz*

*et al., 2013*). Furthermore, *PGBD5* expression in both human and mouse appears largely restricted to the early embryo and certain areas of the embryonic and adult brain (*Sarkar et al., 2003*; *Pavelitz et al., 2013*). These intriguing features prompted us to investigate whether human PGBD5 has retained the enzymatic capability of mobilizing DNA.

## Results

Human PGBD5 contains a C-terminal RNase H-like domain that has approximately 20% sequence identity and 45% similarity to the active lepidopteran piggyBac, ciliate piggyMac, and mammalian piggyBat transposases (*Figure 1*) (*Sarkar et al., 2003*; *Baudry et al., 2009*; *Mitra et al., 2013*). In addition, the human genome contains 2358 sequence elements with similarity to the *piggyBac* transposable elements (*Table 1* and *Figure 2A*). Specifically, MER75 (MER75, MER75A, MER75B) and MER85 elements show considerably similar ITR sequences as compared to lepidopteran *piggyBac* transposons (*Table 2* and *Figure 2B*). A total of 328 *piggyBac*-like elements in the human genome have intact ITRs and exhibit duplications of their presumed TTAA target sites (*Table 1* and *Figure 2C*). We reasoned that even though the ancestral transposon substrates of PGBD5 cannot be predicted due to its very ancient evolutionary origin (∼500 My), preservation of its transposase activity should confer residual ability to mobilize distantly related *piggyBac*-like transposons. To test this hypothesis, we used a synthetic transposon reporter PB-EF1-NEO comprised of a neomycin resistance gene flanked by *T. ni piggyBac* ITRs (*Figure 3B*) (*Cary et al., 1989*; *Fraser et al., 1995*). We transiently transfected human embryonic kidney (HEK) 293 cells, which lack endogenous *PGBD5* expression with the PB-EF1-NEO transposon reporter plasmid in the presence of a plasmid expressing PGBD5, and assessed genomic integration of the reporter using clonogenic assays in the presence of G418 to select cells with genomic integration conferring neomycin resistance (*Figure 3C*, *Figure 3—figure supplement 1*). Given the absence of suitable antibodies to monitor PGBD5 expression, we expressed PGBD5 as an N-terminal fusion with the green fluorescent protein (GFP). We observed significant rates of neomycin resistance of cells conferred by the transposon reporter with GFP-PGBD5, but not in cells expressing control GFP or mutant GFP-PGBD5 lacking the transposase domain (*Figure 3C*), despite equal expression of all transgenes (*Figure 3—figure supplement 2*). The efficiency of neomycin resistance conferred by the transposon reporter with GFP-PGBD5 was approximately 4.5-fold less than that of the *T. ni* piggyBac-derived transposase (*Figure 3D*), consistent with their evolutionary divergence. These results suggest that human PGBD5 can promote genomic integration of a *piggyBac*-like transposon.

If neomycin resistance conferred by the PGBD5 and the transposon reporter is due to genomic integration and DNA transposition, then this should require specific activity on the transposon ITRs. To test this hypothesis, we generated transposon reporters with mutant ITRs and assayed them for genomic integration (*Figure 3B*, *Figure 3—figure supplement 3*). DNA transposition by the piggyBac family transposases involves hairpin intermediates with a conserved 5′-GGGTTAACCC-3′ sequence that is required for target site phosphodiester hydrolysis (*Mitra et al., 2008*). Thus, we generated reporter plasmids lacking ITRs entirely or containing complete ITRs with 5′-ATATTAACCC-3′ mutations predicted to disrupt the formation of productive hairpin intermediates (*Mitra et al., 2008*). To enable precise quantitation of mobilization activity, we developed a quantitative genomic PCR assay using primers specific for the transposon reporter and the endogenous human *TK1* gene for normalization (*Figure 3—figure supplements 4, 5*). In agreement with the results of the clonogenic neomycin resistance assays, we observed efficient genomic integration of the donor transposons in cells transfected by GFP-PGBD5 as compared to the minimal signal observed in cells expressing GFP control (*Figure 3E*). Deletion of transposon ITRs from the reporter reduced genomic integration to background levels (*Figure 3E*). Consistent with the specific function of *piggyBac* family ITRs in genomic transposition, mutation of the terminal GGG sequence in the ITR significantly reduced the integration efficiency (*Figure 3E*). These results indicate that specific transposon ITR sequences are required for PGBD5-mediated DNA transposition.

DNA transposition by piggyBac superfamily transposases is distinguished from most other DNA transposon superfamilies by the precise excision of the transposon from the donor site and preference for insertion in TTAA sites (*Cary et al., 1989*; *Fraser et al., 1995*). To determine the structure of the donor sites of transposon reporters mobilized by PGBD5, we isolated plasmid DNA from cells 2 days after transfection, amplified the transposon reporter using PCR, and determined its sequence using capillary Sanger sequencing (*Figure 4—figure supplement 1*). Similar to the hyperactive *T. ni*

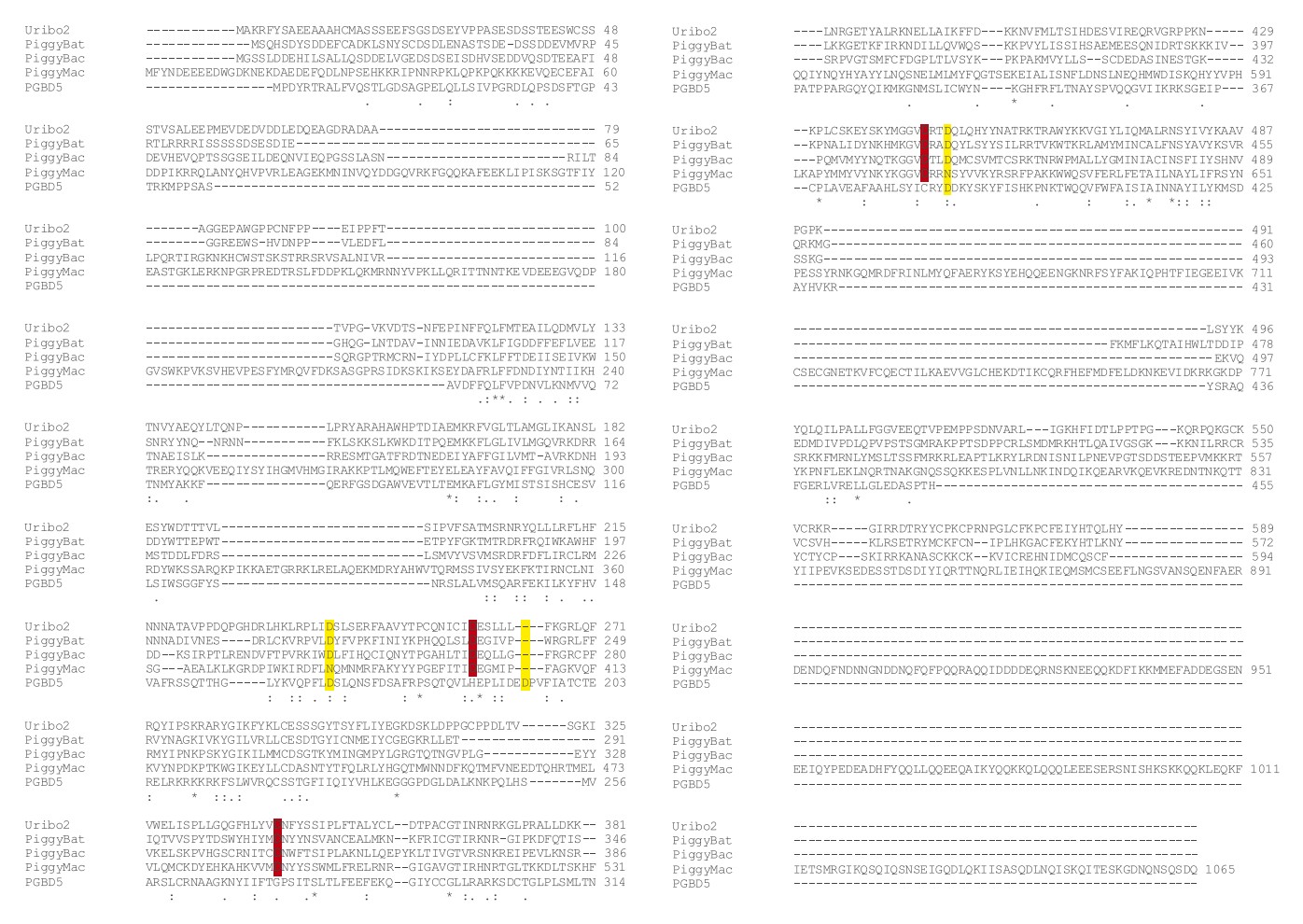

**Figure 1**. Human PGBD5 is a distinct piggyBac-like transposase. Sequence alignment of piggyBac-like transposases frog Uribo 2, bat piggyBat, lepidopteran piggyBac, and human PGBD5. Catalytic residues conserved among piggyBac transposases are highlighted in red. Human PGBD5 D168, D194, and D386 residues, identified in our study (**Figure 7**), are marked in yellow.

piggyBac, cells expressing GFP-PGBD5, but not those expressing GFP control vector, exhibited robust excision of ITR-flanked transposon with apparently precise repair of the donor plasmid (**Figure 4A,B** and **Figure 4—figure supplement 1**). These results suggest that PGBD5 is an active cut-and-paste DNA transposase.

To validate chromosomal integration and determine the location and precise structure of the insertion of the reporter transposons in the human genome, we isolated genomic DNA from G418-resistant HEK293 cells following transfection with PGBD5 and PB-EF1-NEO and amplified the genomic sites of transposon insertions using flanking sequence exponential anchored (FLEA) PCR, a technique originally developed for high-efficiency analysis of retroviral integrations (**Pule et al., 2008**). We adapted FLEA-PCR for the analysis of genomic DNA transposition by using unique reporter sequence to prime polymerase extension upstream of the transposon ITR into the flanking human genome, followed by reverse linear extension using degenerate primers, and exponential amplification using specific nested primers to generate chimeric amplicons suitable for massively parallel single-molecule Illumina DNA sequencing (**Figure 5**) (**Henssen et al., 2015a**). This method enabled us to isolate specific portions of the human genome flanking transposon insertions, as evidenced by the reduced yield of amplicons isolated from control cells lacking transposase vectors or expressing GFP (**Figure 6—figure supplement 1**). To identify the sequences of the transposon genomic insertions at single-base pair resolution, we aligned reads obtained from FLEA-PCR Illumina

**Table 1**. Summary of annotated human piggyBac-like elements

| | Total piggyBac-like elements | Intact elements* |
|---|---|---|
| MER75 | 475 | 144 |
| MER75A | 93 | 62 |
| MER75B | 114 | 27 |
| MER85 | 905 | 95 |
| UCON29 | 240 | 0 |
| Looper | 531 | 0 |

*Denotes elements with intact ITR sequences that align with the consensus without gaps and contain a TTAA target site duplication.

ITR, inverted terminal repeat.

sequencing to the human hg19 reference genome and synthetic transposon reporter, and identified split reads that specifically span both (*Figure 5*). These data have been deposited to the Sequence Read Archive (http://www.ncbi.nlm.nih.gov/sra/, accession number SRP061649, *Henssen et al., 2015c*), with the processed and annotated data available from the Dryad Digital Repository (*Henssen et al., 2015b*).

To infer the mechanism of genomic integration of transposon reporters, we analyzed the sequences of the insertion loci to determine integration preferences at base-pair resolution and identify potential sequence preferences. We found that transposon amplicons isolated from cells expressing GFP-PGBD5, but not those isolated from GFP control cells, were significantly enriched for TTAA sequences, as determined by sequence entropy analysis (*Crooks et al., 2004*) (*Figure 6A*). To discriminate between potential DNA transposition at TTAA target sites and alternative mechanisms of chromosomal integration, we classified genomic insertions based on target sites containing TTAA and those containing other sequence motifs (*Table 3*). Consistent with the DNA transposition activity of PGBD5, we observed significant induction of TTAA-containing insertions in cells expressing GFP-PGBD5 and transposons with intact ITRs, as compared to control cells expressing GFP, or to cells transfected with GFP-PGBD5 and mutant ITR transposons (*Table 3*). Sequence analysis of split reads containing transposon-human junction at TTAA sites revealed that, in almost every case examined ($n$ = 65 out of 66), joining between TTAA host and transposon DNA

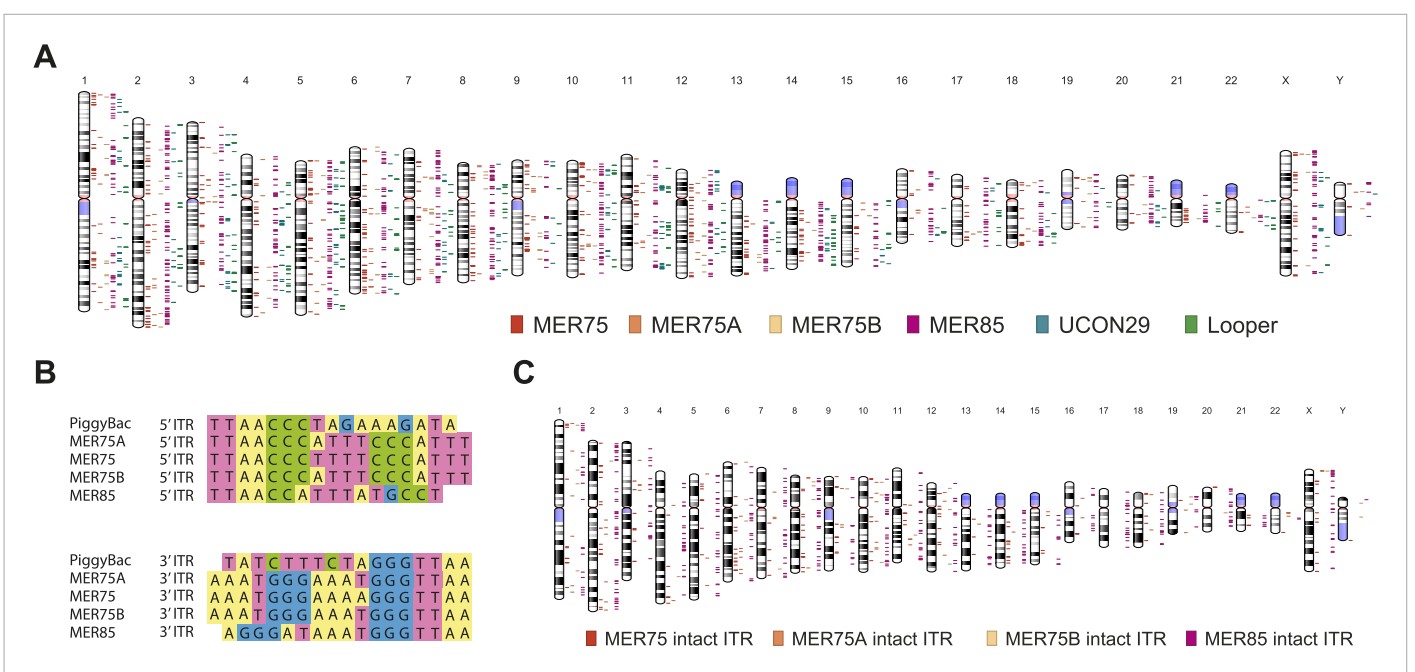

**Figure 2**. Human piggyBac-like transposable elements have intact inverted terminal repeat sequences similar to the *T. ni* piggyBac transposon. (**A**) Chromosome ideogram representing the distribution of annotated piggyBac-like elements in the human genome (version hg19). (**B**) Multiple sequence alignment of the piggybac inverted terminal repeat (ITR) sequence with the consensus ITR sequences of the MER75 and MER85 families of piggyBac-like elements. (**C**) Chromosome ideogram representing the distribution of piggyBac-like elements annotated in the human genome (version hg19) with TTAA target site duplication as well as ITR sequences aligning with the consensus (intact ITRs).

**Table 2.** Sequence identity matrix of the piggy-Bac inverted terminal repeat sequences and consensus sequences of the MER75 and MER85 human piggyBac-like elements

|         | piggyBac | MER75 | MER85 |
|---------|----------|-------|-------|
| piggyBac | 100%    | –     | –     |
| MER75   | 53%      | 100%  | –     |
| MER85   | 56%      | 63%   | 100%  |

occurred precisely at the GGG/CCC terminal motif of the donor transposon ITR (*Figure 6C*), in agreement with its requirement for efficient DNA transposition (*Figure 3E*). Consistent with the genome-wide transposition induced by PGBD5, we identified transposition events in all human chromosomes, including both genic and intergenic loci (*Figure 6B*). Thus, PGBD5 can mediate canonical cut-and-paste DNA transposition of *piggyBac* transposons in human cells.

Requirement for transposon substrates with specific ITRs, their precise excision, and preferential insertion into TTAA-containing genomic locations is all consistent with the preservation of PGBD5's DNA transposase activity in human cells. Like other cut-and-paste transposases, piggyBac superfamily transposases are thought to utilize a triad of aspartate or glutamate residues to catalyze phosphodiester bond hydrolysis, but the catalytic triad of aspartates previously proposed for *T. ni* piggyBac is apparently not conserved in the primary sequence of PGBD5 (*Figure 1*) (*Sarkar et al., 2003*; *Keith et al., 2008*; *Mitra et al., 2008*; *Nesmelova and Hackett, 2010*). Thus, we hypothesized that distinct aspartic or glutamic acid residues may be required for DNA transposition mediated by PGBD5. To test this hypothesis, we used alanine-scanning mutagenesis and assessed transposition activity of GFP-PGBD5 mutants using quantitative genomic PCR (*Figure 7* and *Figure 7—figure supplements 1–3*). This analysis indicated that simultaneous alanine mutations of D168, D194, and D386 reduced apparent transposition activity to background levels, similar to that of GFP control (*Figure 7A*). We confirmed that the mutant GFP-PGBD5 proteins have equivalent stability and expression as the wild-type protein in cells by immunoblotting against GFP (*Figure 7B*). Phylogenetic analysis of vertebrate piggyBac homologs from *Danio rerio*, *Python bivittatus*, *Xenopus tropicalis*, *Gallus gallus*, *Mus musculus*, and *Homo sapiens* showed that PGBD5 proteins are divergent from other piggyBac-like proteins (*Figure 8*) and include conservation of functionally important D168, D194, and D386 residues that distinguish them from lepidopteran piggyBac and human PGBD1-4 (*Figure 8—figure supplement 1*). These results suggest that PGBD5 represents a distinct member of the evolutionarily ancient piggyBac-like family of DNA transposases.

## Discussion

Our current findings indicate that human PGBD5 is an active *piggyBac* transposase that can catalyze DNA transposition in human cells. DNA transposition by PGBD5 requires its C-terminal transposase domain and depends on specific ITRs derived from the lepidopteran *piggyBac* transposons (*Figure 3*). DNA transposition involves transesterification reactions mediated by DNA hairpin intermediates (*Mitra et al., 2008*). Consistent with the requirement of intact termini of the piggyBac, Tn10, and Mu transposons (*Elick et al., 1997*), elimination or mutation of the terminal GGG nucleotides from the transposon substrates also abolishes the transposition activity of PGBD5 (*Figure 3*). PGBD5-induced DNA transposition is precise with preference for insertions at TTAA genomic sites (*Figure 4*). Since our analysis was limited to ectopically expressed PGBD5 fused to GFP and episomal substrates derived from lepidopteran *piggyBac* transposons, it is possible that endogenous PGBD5 may exhibit different activities on chromatinized substrates in the human genome.

Current structure-function analysis indicates that PGBD5 requires three aspartate residues to mediate DNA transposition (*Figure 7*), but its transposase domain appears to be distinct from other piggyBac transposase enzymes with respect to its primary sequence (*Figure 1* and *Figure 8*) (*Keith et al., 2008*). Thus, the three aspartate residues required for efficient DNA transposition by PGBD5 may form a catalytic triad that functions in phosphodiester bond hydrolysis, similar to the DDD motif in other *piggyBac* family transposases, or alternatively may contribute to other steps in the transposition reaction, such as synaptic complex formation, hairpin opening, or strand exchange (*Elick et al., 1997*; *Keith et al., 2008*; *Mitra et al., 2008*). In addition, we find that alanine mutations of the three required aspartate residues in the PGBD5 transposase domain significantly reduce but do not completely eliminate genomic integration of the transposon reporters (*Figure 7*). This could reflect residual catalytic activity despite these mutations, or that PGBD5 expression may affect other mechanisms of DNA integration in human cells.

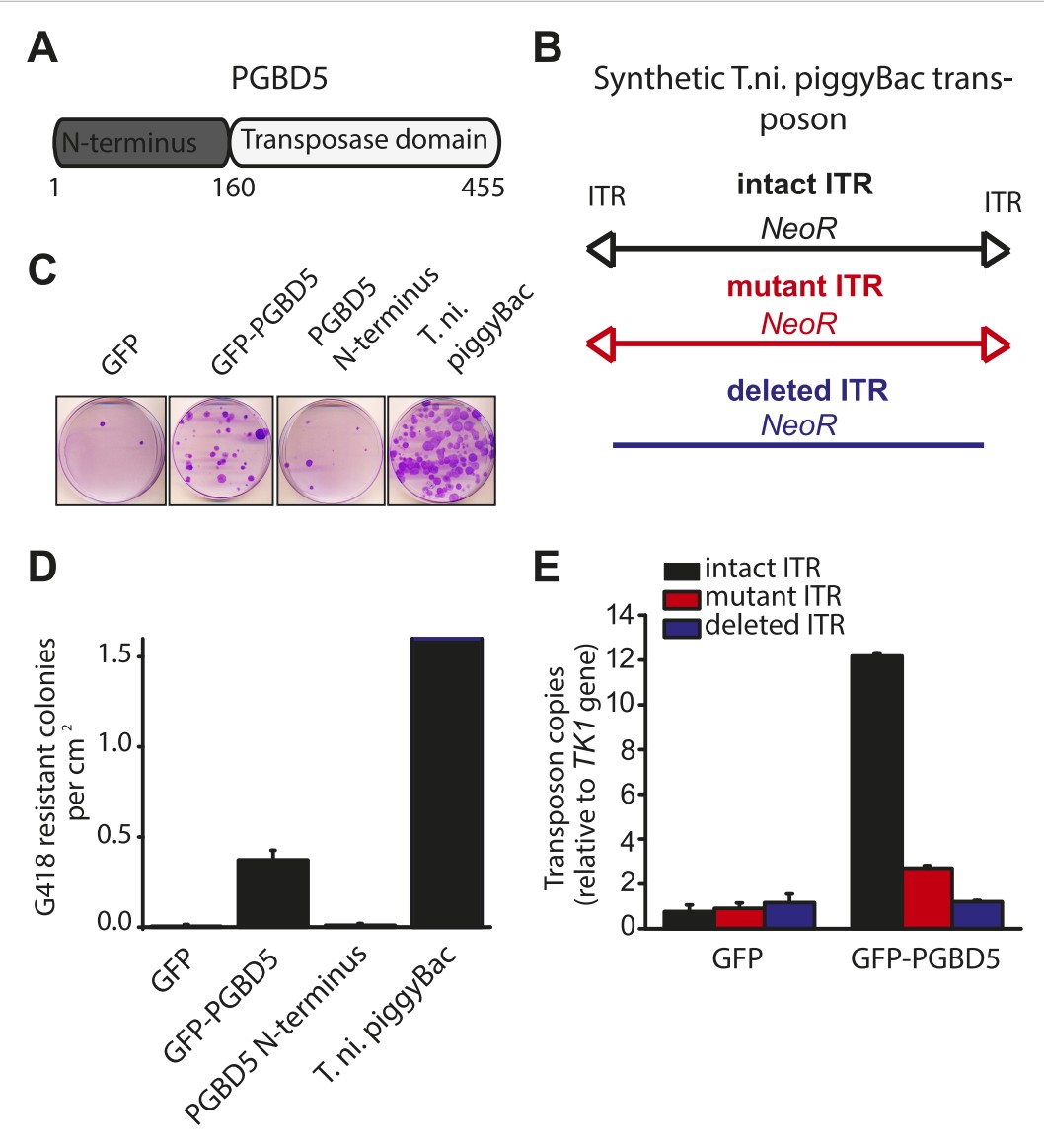

**Figure 3**. PGBD5 induces genomic integration of synthetic piggyBac transposons in human cells. (**A**) Schematic of the human PGBD5 protein with its C-terminal transposase homology domain, as indicated. (**B**) Schematic of synthetic transposon substrates used for DNA transposition assays, including transposons with mutant ITR marked by triangles in red, and transposons lacking ITRs marked in blue. (**C**) Representative photographs of crystal violet-stained colonies obtained after G418 selection of HEK293 cells co-transfected with the transposon reporter plasmid along with transposase cDNA expression vectors. (**D**) Quantification of G418-selection clonogenic assays, demonstrating the integration activities of GFP-PGBD5, PGBD5 N-terminus, *T. ni.* piggyBac, and green fluorescent protein (GFP) control (GFP-PGBD5 vs GFP; p = 0.00031). (**E**) Quantification of genomic transposon integration using quantitative PCR of GFP-PGBD5 and GFP expressing cells using intact (black), mutant (red), and deleted (blue) ITR-containing transposon reporters (intact vs mutant ITR; p = 0.00011). Error bars represent standard errors of the mean of 3 biologic replicates.

The following figure supplements are available for figure 3:

**Figure supplement 1**. Assay for genomic integration of transposon reporters.

**Figure supplement 2**. GFP-PGBD5, PGBD5 N-terminus, and *T. ni.* piggyBac are equally expressed upon transfection in HEK293 cells.

*Figure 3. continued on next page*

*Figure 3. Continued*

**Figure supplement 3**. Sanger sequencing traces of the ITR of the synthetic transposon reporter plasmids.

**Figure supplement 4**. Quantitative assay of genomic integration of transposon reporters.

**Figure supplement 5**. Quantitative genomic PCR standard curve for transposon specific primers.

The evolutionary conservation of the transposition activity of PGBD5 suggests that it may have hitherto unknown biologic functions among vertebrate organisms. DNA transposition is a major source of genetic variation that drives genome evolution, with some DNA transposases becoming extinct and others domesticated to evolve exapted functions. The evolution of transposons' activities can be highly variable, with some organisms such as *Zea mays* undergoing continuous genome remodeling and recent twofold expansion through endogenous retrotransposition, *Drosophila* and *Saccharomyces* owing over half of their known spontaneous mutations to transposons, and primate species including humans exhibiting relative extinction of transposons (*Feschotte and Pritham, 2007*).

Indeed, transposase-derived genes domesticated in humans have evolved to have endogenous functions other than genomic transposition per se. For example, human RAG1 is a domesticated *Transib* transposase that has retained its active transposase domain, and can transpose ITR-containing transposons in vitro, but catalyzes somatic recombination of immunoglobulin and T-cell receptor genes in lymphocytes across signal sequences that might be derived from related transposons (*Landree et al., 1999*; *Fugmann et al., 2000*). Human SETMAR is a *Mariner*-derived transposase with a divergent DDN transposase domain that has retained its endonuclease activity and functions in double-strand DNA repair by non-homologous end joining (*Liu et al., 2007*). The human genome encodes over 40 other genes derived from DNA transposases (*Smit, 1999*; *Feschotte and Pritham, 2007*), including *THAP9* that was recently found to mobilize transposons in human cells with as of yet

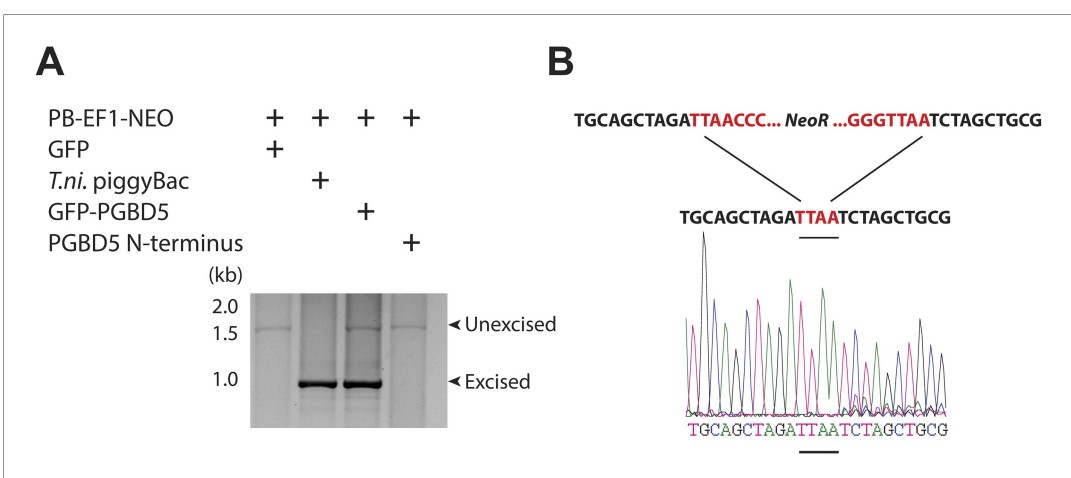

**Figure 4**. PGBD5 precisely excises piggyBac transposons. (**A**) Representative agarose electrophoresis analysis of PCR-amplified PB-EF1-NEO transposon reporter plasmid from transposase-expressing cells, demonstrating efficient excision of the ITR-containing transposon by PGBD5, but not GFP or PGBD5 N-terminus mutant lacking the transposase domain. *T. ni* piggyBac serves as positive control. (**B**) Representative Sanger sequencing fluorogram of the excised transposon, demonstrating precise excision of the ITR and associated duplicated TTAA sequence, marked in red, demonstrating integrations of transposons (green) into human genome (blue) with TTAA insertion sites and genomic coordinates, as marked.

The following figure supplement is available for figure 4:

**Figure supplement 1**. Schematic of transposon excision assay.

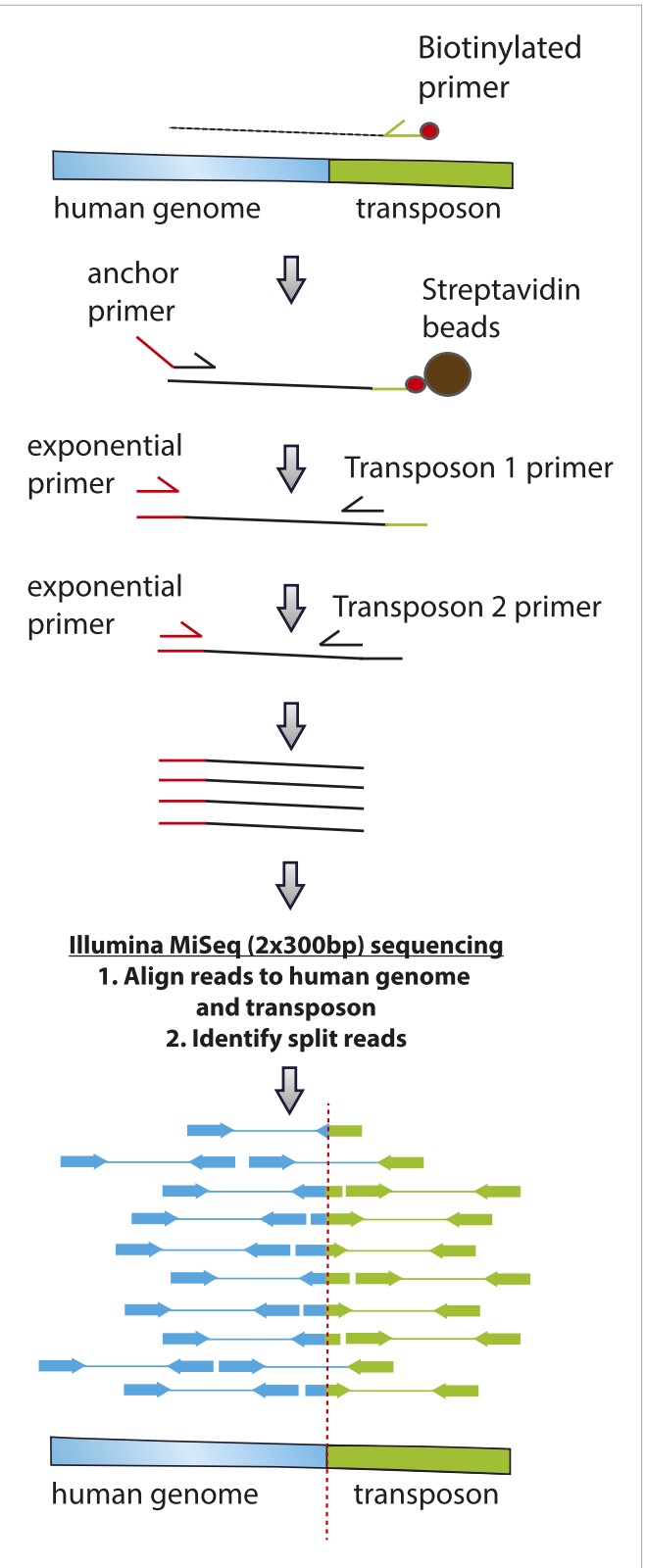

**Figure 5**. Schematic of transposon-specific flanking sequence exponential anchored–polymerase chain reaction amplification (FLEA-PCR) and massively parallel single molecule sequencing assay for mapping and sequencing transposon insertions.

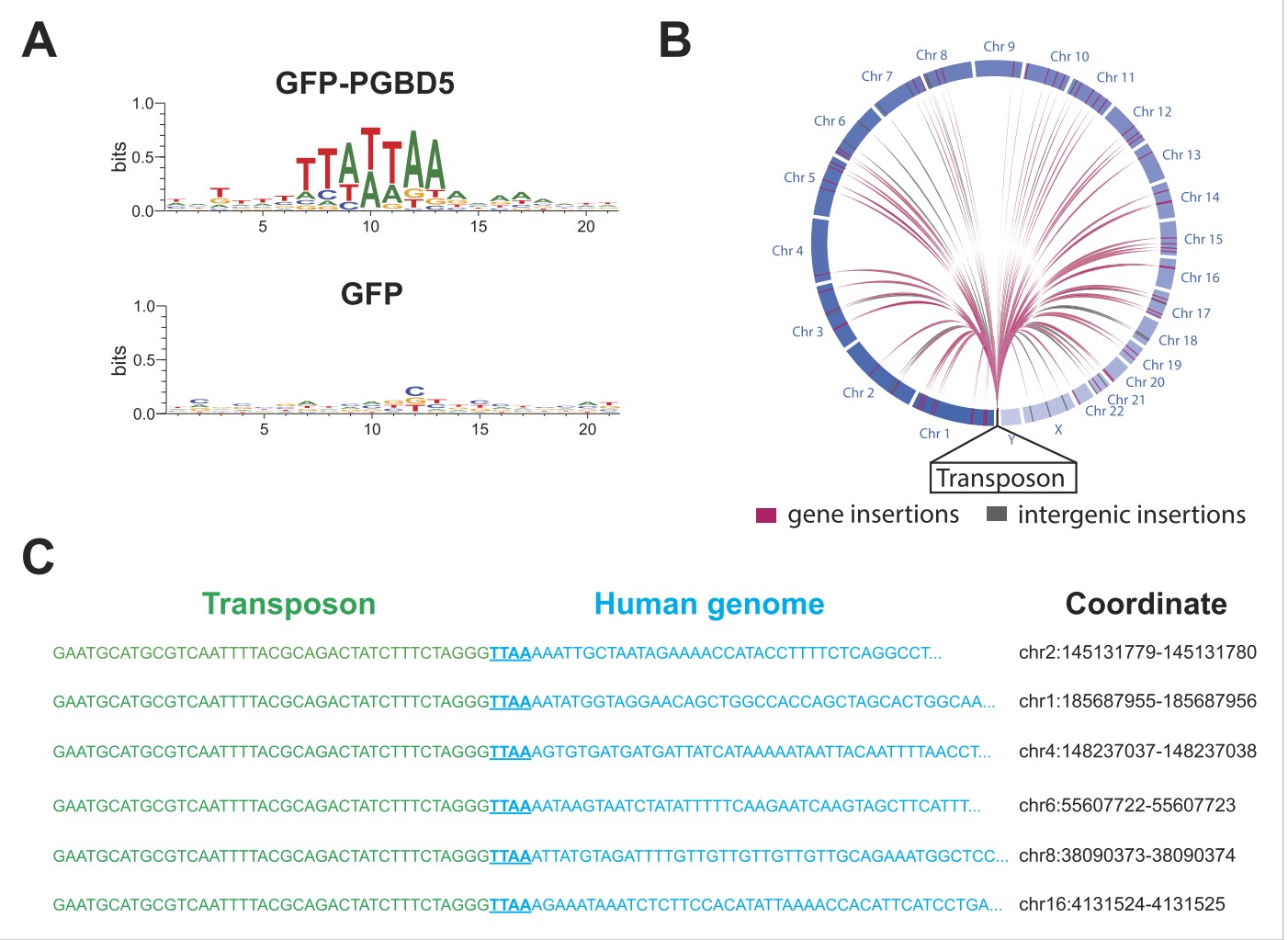

**Figure 6**. PGBD5 induces DNA transposition in human cells. (**A**) Analysis of the transposon integration sequences, demonstrating TTAA preferences in integrations in cells expressing GFP-PGBD5, but not GFP control. X-axis denotes nucleotide sequence logo position, and y-axis denotes information content in bits. (**B**) Circos plot of the genomic locations PGBD5-mobilized transposons plotted as a function of chromosome number and transposition into genes (red) and intergenic regions (gray). (**C**) Alignment of representative DNA sequences of identified genomic integration sites.

The following figure supplement is available for figure 6:

**Figure supplement 1**. Representative agarose gel image of amplicons from flanking sequence exponential anchored–polymerase chain reaction amplification (FLEA-PCR).

unknown function (*Majumdar et al., 2013*). RAG1, THAP9, and PGBD5 are, to our knowledge, the only human proteins with demonstrated transposase activity in human cells.

The distinct biochemical and structural features of PGBD5 indicated by our findings are consistent with its unique evolution and function among human *piggyBac*-derived transposase genes (*Sarkar et al., 2003*; *Pavelitz et al., 2013*). *PGBD5* exhibits deep evolutionary conservation predating the origin of vertebrates, including a preservation of genomic synteny across lancelet, lamprey, teleosts, and amniotes (*Pavelitz et al., 2013*). This suggests that while PGBD5 likely derived from an autonomous mobile element, this ancestral copy was immobilized early in evolution and PGBD5 can probably no longer mobilize its own genomic locus, at least in germ line cells. The human genome contains several thousands of miniature inverted repeat transposable elements (MITEs) with similarity to *piggyBac* transposons (*Figure 2* and *Table 1*) (*Sarkar et al., 2003*; *Feschotte and Pritham, 2007*). CSB-PGBD3 can bind to the *piggyBac*-derived *MER85* elements in

**Table 3**. Analysis of transposon integration sequences in human genomes induced by PGBD5

| | Intact transposon | | Mutant transposon | |
|---|---|---|---|---|
| | **TTAA ITR** | **Non-ITR** | **TTAA ITR** | **Non-ITR** |
| Transposase | | | | |
| GFP-PGBD5 | 82% (65)[†] | 18% (14) | 11% (4)[‡] | 89% (33) |
| GFP Control | 17% (2) | 83% (10) | 40% (27) | 60% (40) |

Cells expressing GFP-PGBD5 and intact transposons exhibit significantly higher frequency of genomic integration as compared to either GFP control, or GFP-PGBD5 with mutant transposons, with 82% (65 out of 79) of sequences demonstrating DNA transposition of ITR transposons into TTAA sites ([†]$p = 1.8 \times 10^{-5}$). Mutation of the transposon ITR significantly reduces ITR-mediated integration, with only 11% (4 out of 37) of sequences ([‡]$p = 0.0016$). Numbers in parentheses denote absolute numbers of identified insertion sites.
GFP, green fluorescent protein; ITR, inverted terminal repeat.

the human genome (*Bailey et al., 2012*; *Gray et al., 2012*). Similarly, it is possible that PGBD5 can act *in trans* to recognize and mobilize one or several related MITEs in the human genome. Recently, single-molecular maps of the human genome have predicted thousands of mobile element insertions, and the activity of PGBD5 or other endogenous transposases may explain some of these novel variants (*Chaisson et al., 2015*; *Pendleton et al., 2015*).

Given that both human RAG1 and ciliate piggyMac domesticated transposases catalyze the elimination of specific genomic DNA sequences (*Hiom et al., 1998*; *Baudry et al., 2009*), it is reasonable to hypothesize that PGBD5's biological function may similarly involve the excision of as of yet unknown ITR-flanked sequences in the human genome or another form of DNA recombination. Since DNA transposition by *piggyBac* family transposases requires substrate chromatin accessibility and DNA repair, we anticipate that additional cellular factors are required for and regulate PGBD5 functions in cells. Likewise, just as RAG1-mediated DNA recombination of immunoglobulin loci is restricted to B lymphocytes, and rearrangements of T-cell receptor genes to T lymphocytes, potential DNA rearrangements mediated by PGBD5 may be restricted to specific cell types and developmental periods.

*PGBD5* localizes to the cell nucleus and is expressed during embryogenesis and neurogenesis, but its physiological function is not yet known (*Pavelitz et al., 2013*). Generation of molecular diversity through DNA recombination during nervous system development has been a long-standing hypothesis (*Dreyer et al., 1967*; *Wu and Maniatis, 1999*). The recent discovery of somatic retrotransposition in human neurons (*Coufal et al., 2009*; *Evrony et al., 2012*; *Upton et al., 2015*), combined with our finding of DNA transposition activity by human PGBD5, which is highly expressed in neurons, suggests that additional mechanisms of somatic genomic diversification may contribute to vertebrate nervous system development.

Because DNA transposition is inherently topological and orientation of transposons can affect the arrangements of reaction products (*Claeys Bouuaert et al., 2011*), potential activities of PGBD5 can depend on the arrangements of accessible genomic substrates, leading to both conservative DNA transposition involving excision and insertion of transposon elements, as well as irreversible reactions such as DNA elimination and chromosomal breakage-fusion-bridge cycles, as originally described by *McClintock (1942)*. Finally, given the potentially mutagenic activity of active DNA transposases, we anticipate that unlicensed activity of PGBD5 and other domesticated transposases can be pathogenic in specific disease states, particularly in cases of aberrant chromatin accessibility, such as cancer.

## Materials and methods

### Reagents

All reagents were obtained from Sigma–Aldrich (St. Louis, MO, United States) if not otherwise specified. Synthetic oligonucleotides were obtained from Eurofins MWG Operon (Huntsville, AL, United States) and purified by HPLC.

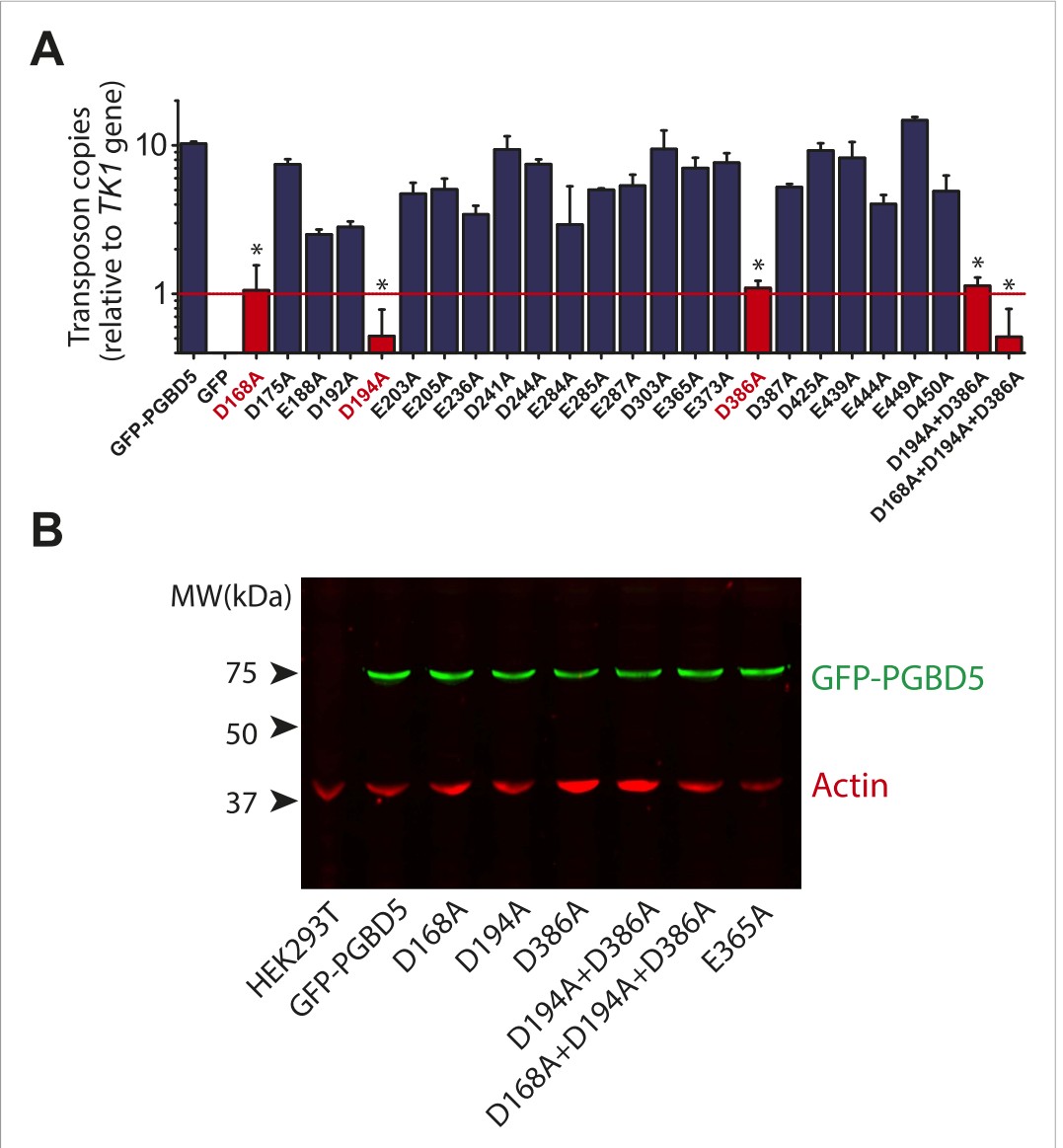

**Figure 7**. Structure-function analysis of PGBD5-induced DNA transposition using alanine scanning mutagenesis. (**A**) Quantitative PCR analysis of genomic integration activity of alanine point mutants of GFP-PGBD5, as compared to wild-type and GFP control-expressing cells. D168A, D194A, and D386A mutants (red) exhibit significant reduction in apparent activity (Asterisks denote statistical significance: p = 0.00011, p = 0.000021, p = 0.000013 vs GFP-PGBD5, respectively). Dotted line marks threshold at which less than 1 transposon copy was detected per haploid human genome. Error bars represent standard errors of the mean of 3 biological replicates. (**B**) Western immunoblot showing equal expression of GFP-PGBD5 mutants, as compared to wild-type GFP-PGBD5 (green). β-actin (red) serves as loading control.

The following figure supplements are available for figure 7:

**Figure supplement 1**. Sanger sequencing trances of pRecLV103-GFP-PGBD5 D>A and E>A mutants (D168A, D175A, E188A, D192A, D194A, E203A, E205A, E236A).

**Figure supplement 2**. Sanger sequencing trances of pRecLV103-GFP-PGBD5 D>A and E>A mutants (D241A, D244A, E284A, E285A, E287A, D303A, E365A, E373A).

**Figure supplement 3**. Sanger sequencing trances of pRecLV103-GFP-PGBD5 D>A and E>A mutants (D386A, D387A, D425A, E439A, E444A, E449A, D450A).

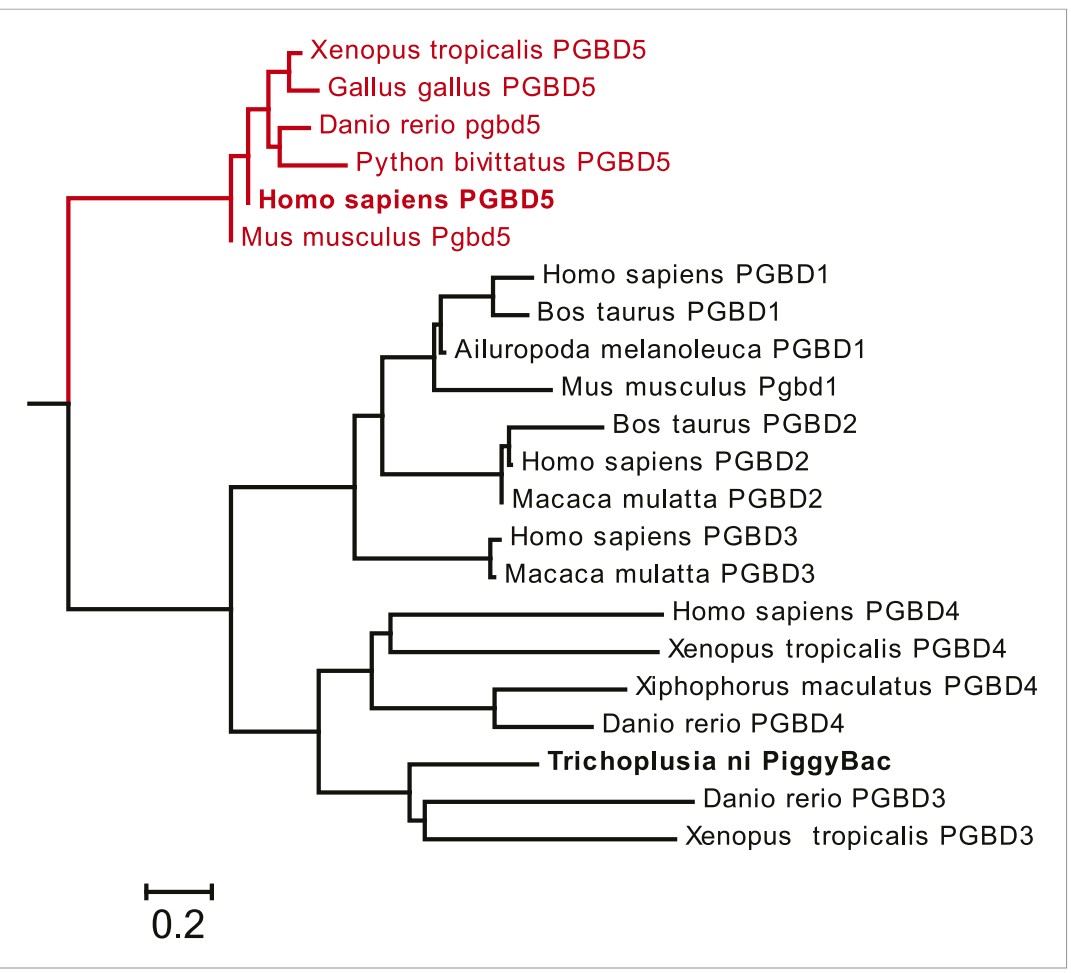

**Figure 8**. *PGBD5* homologs are divergent from other *piggyBac* genes in vertebrates. Phylogenetic reconstruction of the evolutionary relationships among piggyBac transposase-derived genes in vertebrates, demonstrating the PGBD5 homologs represent a distinct subfamily of piggyBac-like derived genes. Scale bar represents Grishin distance.

The following figure supplement is available for figure 8:

**Figure supplement 1**. PGBD5 glutamic acid resitues D168, D194, and D386 are conserved across species.

## Cell culture

HEK293 and HEK293T were obtained from the American Type Culture Collection (ATCC, Manassas, Virginia, United States). The identity of all cell lines was verified by STR analysis and lack of *Mycoplasma* contamination was confirmed by Genetica DNA Laboratories (Burlington, NC, United States). Cell lines were cultured in DMEM supplemented with 10% fetal bovine serum and 100 U/ml penicillin and 100 µg/ml streptomycin in a humidified atmosphere at 37°C and 5% $CO_2$.

## Plasmid constructs

Human PGBD5 cDNA (Refseq ID: NM_024554.3) was cloned as a GFP fusion into the lentiviral vector pReceiver-Lv103-E3156 (GeneCopoeia, Rockville, MD, United States). *piggyBac* ITRs (5′-TTAACCCTAGAAAGATAATCATATTGTGACGTACGTTAAAGATAATCATGTGTAAAATTGACGCATG-3′ and 5′-CATGCGTCAATTTTACGCAGACTATCTTTCTAGGGTTAA-3′), as originally cloned by Malcolm Fraser et al. (*Elick et al., 1997*; *Handler et al., 1998*), were cloned into PB-EF1-NEO to flank IRES-driven neomycin resistance gene, as obtained from System Biosciences (Mountain View, CA, United States). Plasmid encoding the hyperactive *T. ni* piggyBac transposase, as originally generated by

Nancy Craig et al. (*Li et al., 2013*), was obtained from System Biosciences. Site-directed PCR mutagenesis was used to generate mutants of PGBD5 and *piggyBac*, according to manufacturer's instructions (Agilent, Santa Clara, CA, United States). Plasmids were verified by restriction endonuclease mapping and Sanger sequencing and deposited in Addgene. Lentivirus packaging vectors psPAX2 and pMD2.G were obtained from Addgene (*Cudre-Mauroux et al., 2003*).

## Cell transfection

HEK293 cells were seeded at a density of 100,000 cells per well in a 6-well plate and transfected with 2 µg of total plasmid DNA, containing 1 µg of transposon reporter (PB-EF1-NEO or mutants) and 1 µg of transposase cDNA (pRecLV103-GFP-PGBD5 or mutants) using Lipofectamine 2000, according to manufacturer's instructions (Life Technologies, CA, United States). After 24 hr, transfected cells were trypsinized and re-plated for functional assays.

## Quantitative RT-PCR

Upon transfection, cells were cultured for 48 hr and total RNA was isolated using the RNeasy Mini Kit, according to manufacturer's instructions (Qiagen, Venlo, Netherlands). cDNA was synthesized using the SuperScript III First-Strand Synthesis System (Invitrogen, Waltham, MA, United States). Quantitative real-time PCR was performed using the KAPA SYBR FAST PCR polymerase with 20 ng template and 200 nM primers, according to the manufacturer's instructions (Kapa Biosystems, Wilmington, MA, United States). PCR primers are listed in *Supplementary file 1*. Ct values were calculated using ROX normalization using the ViiA 7 software (Applied Biosystems).

## Neomycin resistance colony formation assay

Upon transfection, cells were seeded at a density of 1000 cells per 10-cm dish and selected with G418 sulfate (2 mg/ml) for 2 weeks. Resultant colonies were fixed with methanol and stained with crystal violet.

## Transposon excision assay

Upon transfection, cells were cultured for 48 hr and DNA was isolated using the PureLink Genomic DNA Mini Kit, according to manufacturer's instructions (Life Technologies). Reporter plasmid sequences flanking the neomycin resistance cassette transposons were amplified using hot start PCR with an annealing temperature of 57°C and extension time of 2 min, according to the manufacturer's instructions (New England Biolabs, Beverly, MA, United States) using the Mastercycler Pro thermocycler (Eppendorf, Hamburg, Germany). PCR primers are listed in *Supplementary file 1*. The PCR products were resolved using agarose gel electrophoresis and visualized by ethidium bromide staining. Identified gel bands were extracted using the PureLink Quick Gel Extraction Kit (Invitrogen) and Sanger sequenced to identify excision products.

## Quantitative PCR assay of genomic transposon integration

Upon transfection, cells were selected with puromycin (5 µg/ml) for 2 days to eliminate non-transfected cells. After selection, cells were expanded for 10 days without selection and genomic DNA isolated using PureLink Genomic DNA Mini Kit (Life Technologies). Quantitative real-time PCR was performed using the KAPA SYBR FAST PCR polymerase with 20 ng template and 200 nM primers, according to the manufacturer's instructions (Kapa Biosystems). PCR primers are listed in *Supplementary file 1*. Ct values were calculated using ROX normalization using the ViiA 7 software (Applied Biosystems). We determined the quantitative accuracy of this assay using analysis of serial dilution PB-E1-NEO plasmid as reference (*Figure 3—figure supplement 5*).

## FLEA-PCR

To amplify genomic transposon integration sites, we modified FLEA-PCR (*Pule et al., 2008*), as described in *Figure 5* (*Henssen et al., 2015a*). First, linear extension PCR was performed using 2 µg of genomic DNA and 100 nM biotinylated linear primer using the Platinum HiFidelity PCR mix, according to manufacturer's instructions (Invitrogen Corp.). Linear extension parameters for PCR were: 95°C (45 s), 62°C (45 s), 72°C (3 min) for 30 cycles. Reaction products were purified by diluting the samples in a total volume of 200 µl of nuclease-free water and centrifugation using the Amicon Ultra 0.5 ml 100 K at 12,000×*g* for 10 min at room temperature (EMD Millipore, Billerica, MA, United States) purification. Retentate was bound to streptavidin ferromagnetic

beads on a shaker at room temperature overnight (Dynal, Oslo, Norway). Beads were washed with 40 µl of washing buffer (Kilobase binder kit; Dynal), then water, then 0.1 N NaOH, and finally with water again.

To anneal the anchor primer, washed beads were resuspended in a total volume of 20 µl containing 5 µM anchor primer, 500 nM dNTP, and T7 DNA polymerase buffer (New England Biolabs). Samples were placed in a heating block pre-heated to 85°C and allowed to passively cool to 37°C. Once annealed, 10 units of T7 DNA polymerase (New England Biolabs) was added and the mixtures were incubated for 1 hr at 37°C. Next, the beads were washed five times in water.

To exponentially amplify the purified products, beads were resuspended in a total volume of 50 µl containing 500 nM of exponential and Transposon1 primers and the Platinum HiFidelity PCR mix. PCR was performed with the following parameters: 95°C for 5 min, followed by 35 cycles of 95°C for 45 s, 62°C for 45 s, and 72°C for 3 min. PCR products were purified using the Invitrogen PCR purification kit (Invitrogen Corp.). Second nested PCR was performed using 1/50th of the first exponential PCR product as template using the Platinum HiFidelity PCR with 500 nM of exponential and Transposon2 primers. PCR was performed with the following parameters: 35 cycles of 95°C for 45 s, 62°C for 45 s, and 72°C for 3 min. Final PCR products were purified using the Invitrogen PCR purification kit, according to the manufacturer's instructions (Invitrogen Corp.).

## Sequencing of transposon reporter integration sites

Equimolar amounts of purified FLEA-PCR amplicons were pooled, as measured using fluorometry with the Qubit instrument (Invitrogen) and sized on a 2100 BioAanalyzer instrument (Agilent Technologies). The sequencing library construction was performed using the KAPA Hyper Prep Kit (Kapa Biosystems) and 12 indexed Illumina adaptors from IDT (Coralville, IO, United States), according to the manufacturer's instructions.

After quantification and sizing, libraries were pooled for sequencing on a MiSeq (pooled library input at 10 pM) on a 300/300 paired end run (Illumina, San Diego, CA, United States). An average of 575,565 paired reads were generated per sample. The duplication rate varied between 56 and 87%. Because of the use of FLEA-PCR amplicons for DNA sequencing, preparation of Illumina sequencing libraries is associated with the formation of adapter dimers (*ILLUMINA, 2015*). We used cutadapt to first trim reads to retain bases with quality score >20, then identify reads containing adapter dimers and exclude them from further analyses (parameters -q 20 -b P7 = <P7_index> -B P5 = <P5_index> -discard; where <P7_index> is the P7 primer adapter with the specific barcode for each library, and <P5_index> is the generic P5 adapter sequence: GATCGGAAGAGCGTCGTGTAGGGAAAGAGTG TAGATCTCGGTGGTCGCCGTATCATT) (*Lindgreen, 2012*). Anchor primer sequences were then trimmed from the reads retained using cutadapt (-gGTGGCACGGACTGATCNNNNNN). Filtered and trimmed reads were mapped to a hybrid reference genome consisting of the hg19 full chromosome sequences and the PB-EF1-NEO plasmid sequence using bwa-mem using standard parameters (*Li and Durbin, 2010*). Mapped reads were then analyzed with LUMPY using split read signatures (*Layer et al., 2014*), and insertion loci were identified using the called variants flagged as interchromosomal translocations (BND) between the plasmid sequence and the human genome. Breakpoints were resolved to base-pair accuracy using split read signatures when possible. Insertion loci were taken with 10 flanking base pairs and aligned with MUSCLE to establish consensus sequence (*Layer et al., 2014*). Genomic distribution of insertion loci was plotted using ChromoViz (https://github.com/elzbth/ChromoViz). All analysis scripts are available from Zenodo (*Henaff et al., 2015a*).

## Lentivirus production and cell transduction

Lentivirus production was carried out as described in *Kentsis et al. (2012)*. Briefly, HEK293T cells were transfected using TransIT with 2:1:1 ratio of the pRecLV103 lentiviral vector, and psPAX2 and pMD2.G packaging plasmids, according to manufacturer's instructions (TransIT-LT1, Mirus, Madison, WI, United States). Virus supernatant was collected at 48 and 72 hr post-transfection, pooled, filtered, and stored at −80°C. HEK293T cells were transduced with virus particles at a multiplicity of infection of five in the presence of 8 µg/ml hexadimethrine bromide. Transduced cells were selected for 2 days with puromycin (5 µg/ml).

## Western blotting

To analyze protein expression by Western immunoblotting, 1 million transduced cells were suspended in 80 µl of lysis buffer (4% sodium dodecyl sulfate, 7% glycerol, 1.25% beta-mercaptoethanol, 0.2 mg/ml

Bromophenol Blue, 30 mM Tris-HCl, pH 6.8). Cells suspensions were lysed using Covaris S220 adaptive focused sonicator, according to the manufacturer's instructions (Covaris, Woburn, CA, United States). Lysates were cleared by centrifugation at 16,000×g for 10 min at 4°C. Clarified lysates (30 µl) were resolved using sodium dodecyl sulfate-polyacrylamide gel electrophoresis and electroeluted using the Immobilon FL PVDF membranes (Millipore). Membranes were blocked using the Odyssey Blocking buffer (Li-Cor, Lincoln, Nebraska) and blotted using the mouse and rabbit antibodies against GFP (1:500, clone 4B10) and β-actin (1:5000, clone 13E5), respectively, both obtained from Cell Signaling Technology (Beverly, MA, United States). Blotted membranes were visualized using goat secondary antibodies conjugated to IRDye 800CW or IRDye 680RD and the Odyssey CLx fluorescence scanner, according to manufacturer's instructions (Li-Cor).

## Multiple sequence alignment analysis of DNA and protein sequences

The transposon annotation of the human genome (assembly hg19) was downloaded from the UCSC website (http://hgdownload.soe.ucsc.edu/goldenPath/hg19/database/rmsk.txt.gz) and converted to the GFF3 annotation format. The sequences of the elements in the piggyBac-like MER75, MER75A, MER75B, MER85, UCON29, and LOOPER families were extracted with 50 flanking base pairs using fastaFromBed from the BedTools genome analysis suite (http://bedtools.readthedocs.org). The set of sequences for each family was aligned using MUSCLE using standard parameters (*Edgar, 2004*). ITR sequences for each family were defined as terminal sequences conserved amongst all family members measured using multiple sequence alignments. We used a cutoff of 70% similarity to determine the fist position of the ITR in the alignment. The multiple sequence alignments were then manually curated to identify the set of 'intact' elements defined by containing both the TTAA target site duplication as well as both 3′ and 5′ ITRs aligning without gaps with the consensus ITR sequence. Multiple sequence alignment of the ITR sequences was also performed with MUSCLE, and the sequence identity matrix calculated using SIAS (http://imed.med.ucm.es/Tools/sias.html), with the following measure of identity:

$$Identity = 100 * \left( \frac{Number\ of\ Identical\ Residues}{Length\ of\ shortest\ sequence} \right).$$

Chromosome ideograms were made using the NCBI's Genome Decoration Tool (http://www.ncbi.nlm.nih.gov/genome/tools/gdp/). Multiple sequence alignment of protein sequences was done using Clustal Omega (*Thompson et al., 1994*; *Sievers and Higgins, 2014*). Pairwise BLAST alignments-based Grishin's sequence distance analysis was done using BLAST (http://blast.ncbi.nlm.nih.gov/) and MEGA6 using standard parameters (*Tamura et al., 2013*).

## Statistical analysis

Statistical significance values were determined using two-tailed non-parametric Mann–Whitney tests for continuous variables, and two-tailed Fisher exact test for discrete variables.

## Acknowledgements

We are grateful to Alejandro Gutierrez, Marc Mansour, Thomas Look, Charles Roberts, Daniel Bauer, Leo Wang, Hao Zhu, and Michael Kharas for critical discussions, Nahum Meller for cloning assistance, and Alan Chramiec for technical support. This work was supported by the University of Essen Pediatric Oncology Research Program (AH), NIH K08 CA160660, Burroughs Wellcome Fund, CureSearch for Children's Cancer, and Hyundai Hope on Wheels (AK), and the Irma T Hirschl and Monique Weill-Caulier Charitable Trusts, the STARR Consortium, the Bert L and N Kuggie Vallee Foundation, and the WorldQuant Foundation (CEM). We thank the MSKCC Integrated Genomics Core Facility and Bioinformatics Core Facility for assistance with DNA sequencing and analysis (NIH P30 CA008748).

## Additional information

### Funding

| Funder | Grant reference | Author |
| --- | --- | --- |
| National Cancer Institute (NCI) | CA160660 | Alex Kentsis |

| Funder | Grant reference | Author |
|---|---|---|
| Burroughs Wellcome Fund (BWF) | | Alex Kentsis |
| CureSearch for Children's Cancer | | Alex Kentsis |
| Hyundai Hope On Wheels (Hope On Wheels) | | Alex Kentsis |
| Irma T. Hirschl Trust (Irma T. Hirschl Charitable Trust) | | Christopher E Mason |
| Vallee Foundation (Bert L. & N. Kuggie Vallee Foundation) | | Christopher E Mason |
| National Cancer Institute (NCI) | CA008748 | Alex Kentsis |

The funders had no role in study design, data collection and interpretation, or the decision to submit the work for publication.

### Author contributions

AGH, AK, Conception and design, Acquisition of data, Analysis and interpretation of data, Drafting or revising the article; EH, CF, CEM, Analysis and interpretation of data, Drafting or revising the article; EJ, ARE, JRC, CMV, MR, ES, JG, Acquisition of data, Drafting or revising the article; MB, Acquisition of data, Analysis and interpretation of data, Drafting or revising the article

### Author ORCIDs

Elizabeth Henaff, http://orcid.org/0000-0001-7906-9681
Alex Kentsis, http://orcid.org/0000-0002-8063-9191

## Additional files

### Supplementary file

• Supplementary file 1. Sequences of PCR primers.

### Major datasets

The following datasets were generated:

| Author(s) | Year | Dataset title | Dataset ID and/or URL | Database, license, and accessibility information |
|---|---|---|---|---|
| Henssen A, Henaff E, Jiang E, Eisenberg AR, Carson JR, Villasante CM, Ray M, Still E, Burns M, Gandara J, Feschotte C, Mason CE, Kentsis A | 2015 | DNA sequencing | http://www.ncbi.nlm.nih.gov/sra/SRP061649 | Publicly available at the NCBI Short Read Archive (Accession no: SRP061649). |
| Henssen A, Henaff E, Jiang E, Eisenberg AR, Carson JR, Villasante C, Ray M, Still E, Burns M, Gandara J, Feschotte C, Mason CE, Kentsis A | 2015 | Data from: Genomic DNA transposition induced by human PGBD5 | http://datadryad.org/resource/doi:10.5061/dryad.b2hc1 | Available at Dryad Digital Repository under a CC0 Public Domain Dedication. |
| Henaff E, Henssen A, Mason CE, Kentsis A | 2015 | Genomic DNA transposition induced by human PGBD5 (Accompanying Scripts for Computational Analyses) | http://dx.doi.org/10.5281/zenodo.22206 | Publicly available at the Zenodo (Accession no: 10.5281/zenodo.22206). |

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
