## [Decision Letter]

Thank you for submitting your work entitled “Genomic DNA transposition induced by human PGBD5” for peer review at *eLife*. Your submission has been evaluated by James Manley (Senior Editor), a Reviewing Editor, and two reviewers. One of the two reviewers, Harmit Malik, has agreed to share his identity.

The reviewers have discussed the reviews with one another and the Senior Editor has drafted this decision to help you prepare a revised submission.

In this well-performed study, the authors reveal the rather surprising conservation of catalytic activity of PGBD5, a *piggyBac*-derived domesticated transposase gene that is one of the oldest conserved domestication events in vertebrates. One of the reviewers commented that: “this is definitely a result of high importance in the field. Not only is it surprising but it opens the possibility of as yet unanticipated DNA transposition in vertebrates that is itself of high interest.” However, while the results are interesting, there are a number of questions that need to be addressed before the paper will be suitable for publication in *eLife*.

Essential revisions:

1) What and where are the sequences in the human genome that the authors indicate are “*piggyBac* transposon-like”? The authors imply that the human *piggyBac* transposons are not identifiable based on an evolutionary argument, but given the conservation of the TIRs of the *Trichoplusia ni* transposon, how were putative human *piggyBac* transposons attempted to be found? How similar are the terminal and sub-terminal inverted repeats in human DNA elements thought to be related to *piggyBac* elements relative to the *Trichoplusia ni piggyBac* elements?

2) PGBD5 is revealed to have a rather non-canonical catayltic triad of aspartate residues compared to the original well characterized *piggyBacs*. Rather surprisingly, even the dependence on these newly identified aspartates is not obligatory for catalytic activity. The authors (we think correctly) interpret this to imply a rather divergent *piggyBac* lineage gave rise to PGBD5 – however, there is no attempt to identify such lineages from sequence databases and use them to refine ideas about catalytic site conservation. The data presented is quite convincing about the lack of conservation of canonical aspartates, but rather sparse on the evolutionary history and conservation of PGBD5-like domains. This should not be a difficult task and we would ask you to include phylogenetic analysis and multiple alignments to augment the alanine scanning results presented.

3) The current title of the paper should be revised to indicate that the human PGBD5 protein is mobilizing the heterologous *piggyBac* transposon DNA element from *Trichoplusia ni*.

Other issues to address:

1) Given that conservation of the terminal inverted repeats in *the Trichoplusia ni piggyBac* transposon and the statement in the paper that there are human DNA sequences related to *piggybac* transposons in the human genome, have the author tried to see if PGBD5 can mobilize the human sequences?

2) The authors say that the catalytic DDD triad of residues in PGBD5 was not easy to find. In fact, a previous paper from the Weiner group claimed, using phylogenetic evidence, that the PGBD5 protein would be inactive. The alignment shown indicates a lack of acidic residues in common other *PiggyBac* transposases. Did the authors look at the paper from Susan Wessler to see if their catalytic residues coincide with their extensive phylogenetic predictions (The catalytic domain of all eukaryotic cut-and-paste transposase superfamilies. Yuan YW, Wessler SR. Proc Natl Acad Sci U S A. 2011 May 10;108(19):7884–9)?

3) Based on the mutagenesis performed in the paper, are the three D residues identified in the present work found in the protein sequences of any of the other related human PGBD1-4 proteins or in any of the primate-specific *PiggyBac* family members? The authors might indicate, on the supplemental figure showing the canonical catalytic residues for the insect *PiggyBac* transposases, where the three DDD residues found by the alanine mutagenesis shown in this paper.

4) The assays shown indicate that the human PGBD5 protein is less active than the cognate *Trichoplusia ni* transposase in mobilizing the *piggyBac* transposon. Can the authors give a better quantitation of these data and state that explicitly in the text? Based on the figure, it looks like ∼ 4–fold less active.

5) What is the biological consequence of the PGBD5 mediated transposition? We believe this is outside the scope of the study, but could be presented with additional information about expression patterns of this gene, for instance in mouse development.

---

## [Author Response]

Essential revisions:

*1) What and where are the sequences in the human genome that the authors indicate are “*piggyBac *transposon-like”? The authors imply that the human* piggyBac *transposons are not identifiable based on an evolutionary argument, but given the conservation of the TIRs of the* Trichoplusia ni *transposon, how were putative human* piggyBac *transposons attempted to be found? How similar are the terminal and sub-terminal inverted repeats in human DNA elements thought to be related to* piggyBac *elements relative to the* Trichoplusia ni piggyBac *elements?*

*piggyBac*-like sequences in the human genome have previously been annotated in RepBase. We have now extended this annotation by direct analysis of *piggyBac*-like *MER75*, *MER75a*, *MER75b*, *MER85*, *UCON29* and *Looper* sequences. These results indicate that human *piggyBac*-like sequences are present across the entire genome (revised Figure 2), and include a subset of elements with intact inverted terminal repeats and TTAA target site duplications (revised Figure 2 and revised Table 1). While the human *piggyBac*-like elements exhibit sufficient similarity in their inverted terminal repeats to the lepidopteran *piggyBac* transposon to permit these analyses, their sequences are apparently too divergent to enable identification of sub-terminal repeats and sequence elements by pair-wise sequence comparisons alone. The manuscript text has been revised accordingly (Results, first paragraph).

*2) PGBD5 is revealed to have a rather non-canonical catayltic triad of aspartate residues compared to the original well characterized* piggyBacs*. Rather surprisingly, even the dependence on these newly identified aspartates is not obligatory for catalytic activity. The authors (we think correctly) interpret this to imply a rather divergent* piggyBac *lineage gave rise to PGBD5 – however, there is no attempt to identify such lineages from sequence databases and use them to refine ideas about catalytic site conservation. The data presented is quite convincing about the lack of conservation of canonical aspartates, but rather sparse on the evolutionary history and conservation of PGBD5-like domains. This should not be a difficult task and we would ask you to include phylogenetic analysis and multiple alignments to augment the alanine scanning results presented.*

As suggested, we have now compared the protein sequences of vertebrate *piggyBac*-derived genes. This phylogenetic analysis indicates that zebrafish, python, frog, chicken, mouse and human PGBD5 homologs constitute a divergent subfamily of *piggyBac*-like transposases among vertebrates (revised Figure 8). Consistent with our findings of the functional requirement of D168, D194, and D386 residues for PGBD5’s DNA transposition activity, these three residues are conserved among all examined vertebrate PGBD5 homologs, but not among other *piggyBac*-derived genes such as human PGBD1-4 and lepidopteran *piggyBac* (revised Figure 8). We have revised the manuscript text accordingly (Results, last paragraph).

*3) The current title of the paper should be revised to indicate that the human PGBD5 protein is mobilizing the heterologous* piggyBac *transposon DNA element from* Trichoplusia ni*.*

We firmly believe in reporting research findings accurately and without hyperbole. For this reason we made sure to explicitly specify the observed activity of PGBD5 in the Abstract, and note that PGBD5 acts on “DNA sequences containing inverted terminal repeats with similarity to *piggyBac* transposons.” We struggled with devising a title that would comprehensively express the manuscript’s findings while remaining succinct. We considered the following alternatives but found that they are either awkward and cumbersome (*“*Human PGBD5 is a DNA transposase that can mobilize heterologous *piggyBac transposons* into the human genome*”*), or do not sufficiently communicate our findings (“The human PGBD5 domesticated transposase is catalytically active”). We believe that the current title reporting that human PGBD5 can induce transposition into the human genome is the best solution to this need.

Other issues to address:

*1) Given that conservation of the terminal inverted repeats in the* Trichoplusia ni piggyBac *transposon and the statement in the paper that there are human DNA sequences related to* piggybac *transposons in the human genome, have the author tried to see if PGBD5 can mobilize the human sequences?*

In spite of extensive efforts to identify endogenous *piggyBac*-like elements that may serve as substrates for PGBD5, we have not been able to do so using sequence analysis given their extensive divergence in the human genome (revised Figure 2 and Tables 1 and 2). We have considered a pooled library screen to identify potential substrates based on transposition efficiency, but have decided that this would be beyond the scope of the current manuscript.

*2) The authors say that the catalytic DDD triad of residues in PGBD5 was not easy to find. In fact, a previous paper from the Weiner group claimed, using phylogenetic evidence, that the PGBD5 protein would be inactive. The alignment shown indicates a lack of acidic residues in common other* PiggyBac *transposases. Did the authors look at the paper from Susan Wessler to see if their catalytic residues coincide with their extensive phylogenetic predictions (The catalytic domain of all eukaryotic cut-and-paste transposase superfamilies. Yuan YW, Wessler SR. Proc Natl Acad Sci U S A. 2011 May 10;108(19):7884–9)?*

We have now carried out additional phylogenetic analysis of vertebrate *piggyBac*-derived genes, demonstrating conservation of the functionally required D168, D194, and D386 residues among all examined PGBD5 homologs in vertebrates, and their phylogenetic divergence from other *piggyBac*-like transposases (see response to comment #2 above). As pointed out by the referees, Yuan and Wessler published an extensive sequence analysis of eukaryotic transposase-derived genes, including lepidopteran *piggyBac*, but did not examine the vertebrate PGBD5 specifically.

*3) Based on the mutagenesis performed in the paper, are the three D residues identified in the present work found in the protein sequences of any of the other related human PGBD1-4 proteins or in any of the primate-specific* PiggyBac *family members? The authors might indicate, on the supplemental figure showing the canonical catalytic residues for the insect* PiggyBac *transposases, where the three DDD residues found by the alanine mutagenesis shown in this paper.*

We have revised Figure 1 and include new Figure 8 to highlight these residues, as compared to other *piggyBac*-like transposase-derived genes.

*4) The assays shown indicate that the human PGBD5 protein is less active than the cognate* Trichoplusia ni *transposase in mobilizing the* piggyBac *transposon. Can the authors give a better quantitation of these data and state that explicitly in the text? Based on the figure, it looks like ∼ 4*–*fold less active.*

We compared the apparent activity of human PGBD5 to the hyperactive version of the lepidopteran *piggyBac*: “The efficiency of neomycin resistance conferred by the transposon reporter with GFP-PGBD5 was approximately 4.5–fold less than that of the *T. ni* piggyBac-derived transposase” (Results, first paragraph).

5) What is the biological consequence of the PGBD5 mediated transposition? We believe this is outside the scope of the study, but could be presented with additional information about expression patterns of this gene, for instance in mouse development.

We share the referees’ interest in this question, and are actively investigating the physiologic and pathogenic functions of PGBD5. The manuscript includes a discussion of our considerations of this problem (Discussion, paragraphs five to eight).